# in2IN: Leveraging individual Information to Generate Human INteractions

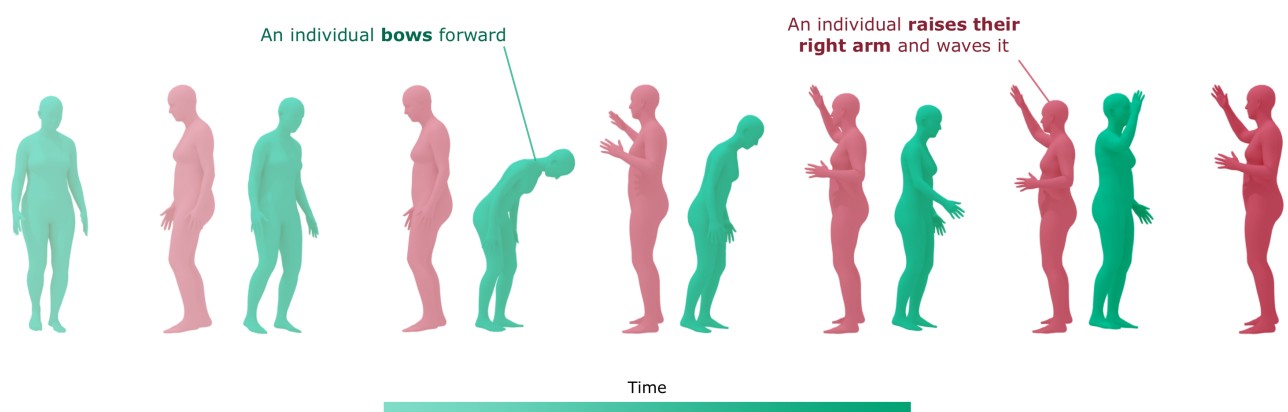

Figure 1. We present in2IN, a diffusion model architecture capable of generating human-human motion interactions using general interaction descriptions to model the inter-personal dynamics and specific individual descriptions to model the intra-personal dynamics. Furthermore, we propose DualMDM, a motion composition method that is able to combine predictions made by an interaction model and by a single-person motion prior, thus increasing the intra-personal diversity of human motion interactions.

## Abstract

Generating human-human motion interactions conditioned on textual descriptions is a very useful application in many areas such as robotics, gaming, animation, and the metaverse. Alongside this utility also comes a great difficulty in modeling the highly dimensional inter-personal dynamics. In addition, properly capturing the intra-personal diversity of interactions has a lot of challenges. Current methods generate interactions with limited diversity of intra-person dynamics due to the limitations of the available datasets and conditioning strategies. For this, we introduce in2IN, a novel diffusion model for human-human motion generation which is conditioned not only on the textual description of the overall interaction but also on the individual descriptions of the actions performed by each person involved in the interaction. To train this model, we use a large language model to extend the InterHuman dataset with individual descriptions. As a result, in2IN achieves state-of-the-art performance in the InterHuman dataset. Furthermore, in order to increase the intra-personal diversity on the existing interaction datasets, we propose DualMDM, a model composition technique that combines the motions generated with in2IN and the motions generated by a single-person motion prior pre-trained on HumanML3D. As a result, DualMDM generates motions with higher individual diversity and improves control over the intra-person dynamics while maintaining inter-personal coherence.

## 1. Introduction

Human Motion Generation refers to creating synthetic human movements that closely mimic those performed by actual individuals. This field has experienced significant advancements alongside the general progress in generative AI over recent years [56]. However, unlike other areas of generative AI, such as image and text generation, annotated motion datasets are scarce due to the need for expensive recording setups and actors. Controlling the generation of a motion based on a given condition is extremely im-

portant for applications such as video games or robotics. We can find many different condition types such as actions [12, 16, 32, 41], audio [27, 43, 47, 55], or natural text [1, 12, 18–20, 25, 26, 33, 34, 40, 41, 48–51, 54]. In contrast to discrete conditioning means such as actions, utilizing text is advantageous due to its capacity to convey detailed descriptions of specific motions. Natural text allows for the specification of movements in different body parts, at varying velocities, and within diverse contexts or emotional states. Recent advancements with Large Language Models (LLMs) have underscored the potency of text as a versatile tool across various applications [10, 14, 42, 53].

Generating realistic individual human motion conditioned on a textual description is a very challenging task due to the complexity of the intra-personal dynamics as well as the difficulty of aligning a textual description with a specific motion. Additionally, motion is rarely done in isolation in the real world. As an intelligent species, we adapt our motions depending on several factors, such as the environment and other individuals that we might interact with [5, 13]. Modeling such interactions is extremely difficult due to the intricacy of inter-personal dynamics [6, 21, 57]. More specifically, a single person might behave in many different ways under the same interaction. This *individual diversity* can arise from variations in the joints trajectories, velocities, or even the action semantics. For example, two people can salute each other by waving the left or the right hand, slowly or quickly, or even bowing instead. Controlling such intra-personal dynamics when generating human-human interactions is an important and underexplored capability. Available annotated interaction datasets such as InterHuman [28] contain a significant amount of annotated interactions. However, neither of them [28, 36, 39] provides enough individual diversity nor detailed textual descriptions of the individual motions of the interaction. As a consequence, recent human-human interaction generation methods [11, 28, 36, 39] tend to replicate the interactions from the training datasets, showing limited diversity in the individual motions that encompass the interactions, and lack individual control capabilities. To address all these problems, we could scale up by collecting bigger and more diverse datasets. This work, instead, proposes a new methodology that effectively exploits the individual diversity already present in the available datasets to improve the performance and control when generating human-human interactions. More particularly, our main contributions[1] are:

- We propose in2IN, a novel diffusion model architecture that is not only conditioned on the overall interaction description but also on the descriptions of the individual motion performed by each interactant, as illustrated in Fig. 1. To do so, we extend the InterHuman dataset [28]

---

[1]The code, model checkpoints, and data will be publically released on: **censored**

with LLM-generated textual descriptions of the individual human motions involved in the interaction. Our approach allows for a more precise interaction generation and achieves state-of-the-art results on the InterHuman dataset.
- We introduce a diffusion conditioning technique based on the Classifier Free Guidance (CFG) [22] that allows weighting independently the importance of each condition during the interaction generation. This enables a higher control over the influence of individual and interaction descriptions on the sampling process.
- We propose DualMDM, a new motion composition technique to further increase the individual diversity and control. By combining our in2IN interaction model with a single-person (individual) motion prior, we generate interactions with more diverse intra-personal dynamics.

## 2. Related Work

### 2.1. Text-Driven Human Motion Generation

A review of recent literature [56] reveals significant progress in this domain over the past two years, with a plethora of methodologies being explored. The first set of methodologies that have been explored is based on aligning the latent spaces of text and motion using the Kullback-Leibler divergence loss [1, 18, 33, 40]. A decoder is trained to convert the text latent representation into the corresponding motion. The main limitation of these approaches is that the scarcity of motion data might lead to latent space misalignments and therefore semantic mismatches between the text and the generated motion.

Based on the recent success of auto-regressive approaches in domains like language, with the advent of LLMs [10, 14, 42, 53] powered by Transformers [44], new approaches have emerged in the motion field [19, 25, 49, 54]. In these, motions are tokenized into discrete codes from a learned codebook, and a Transformer architecture is used to convert text tokens into motion tokens in an autoregressive manner. While these approaches generate more realistic motions, they have some downsides. Firstly, while tokenizing text is a relatively simple task, tokenizing motion is not straightforward because there are no clear individual logic units as can be the words or lemmas in a text. Additionally, due to the nature of auto-regressive models, they cannot model bi-directional dependencies. MMM [34] and MoMask [20] address this limitation using masked attention in BERT [14] style.

Diffusion Models [23, 37] have emerged as the best option for many generative tasks [46], also achieving excellent results in the text-to-motion field. FLAME [26] and MotionDiffusion [51] employ a traditional diffusion model with a Transformer as the noise predictor, achieving state-of-the-art results. Instead of predicting the noise, MDM

[41] predicts the fully denoised motion at each step. This strategy, typically called $x_0$ reparametrization [7, 45], enables the use of kinematic loss functions, leading to better human motion generation. Other methods propose incorporating physical constraints into the diffusion process [48], using latent diffusion models for speeding up the sampling [12], or leveraging retrieval-based methods [50]. Although the sequential multi-step nature of diffusion models during inference makes them very slow, it also empowers them to generate very realistic samples with high diversity [15] and fine-grained control capabilities. As a result, diffusion models are very powerful for human interaction generation.

### 2.2. Text-Driven Human Interaction Generation

ComMDM [36] extends MDM's capabilities to generate multi-human interactions. ComMDM is a cross-attention module integrated into specific layers of the denoisers in two frozen MDM models. This module processes the activations from the two models and adjusts them to foster interaction. In [39], a similar concept is employed but this time with two distinct models. Interaction modeling is achieved through a shared cross-attention module that connects both models, an architecture particularly suited for asymmetric interactions involving an actor and a receiver. However, they observed that their method overfitted to the training dataset due to the lack of annotated interaction datasets. Recently, InterHuman [28] was released, becoming the most extensive annotated dataset of human interactions up to date. The authors also propose a baseline method called InterGen, which is based on two cooperative denoisers with shared weights. Finally, MoMat-MoGen [11] extends the retrieval diffusion model proposed in [50] and adapts it to human interactions, becoming the current state of the art on InterHuman. In contrast to the previous approaches, we propose a diffusion model (in2IN) that conditions the generation on both the general interaction description and a fine-grained description providing more details on the action performed by each individual involved in the interaction. This results in a model that generates adequate inter-personal dynamics and, at the same time, enables precise control on the intra-personal dynamics.

### 2.3. Human Motion Composition

The iterative paradigm underlying diffusion models provides them the capability to combine data, such as multiple images or motions, in a harmonized way [4, 52]. In the realm of motion, the literature has traditionally differentiated between temporal and spatial composition. Temporal composition refers to combining multiple individual motions into a larger sequence [2, 8, 36], making smooth and realistic transitions among them emerge. On the other hand, spatial composition refers to combining multiple motions to generate a new motion of the same length that combines certain elements of the original motions, such as the actions, the trajectory, or joint-specific movements [3, 40]. All of them share the same limitation though: they apply to single-person motion composition. In a more broad sense, [36] proposed a generic *model composition* technique to combine the sampling processes of two different diffusion models, thus generating a harmonized motion. However, they used a fixed score-merging technique along the whole denoising process, which we prove is a suboptimal strategy in more complex scenarios like ours. Instead, we propose a novel model composition technique (DualMDM) that can combine 1) individual motions generated with a prior pre-trained on a single-person motion dataset, and 2) the interactive motions generated by a human-human interaction model like in2IN. The interactions generated with DualMDM show higher diversity of intra-personal dynamics while still maintaining the inter-personal coherence of the overall interaction.

## 3. Method

In this section, we introduce our main methodological contributions. First, in Sec. 3.1, we describe in2IN, our proposed interaction-aware diffusion model conditioned on both the interaction and the individual textual descriptions. Then, we introduce the multi-weight CFG technique, which increases the user control over the influence that each condition has over the generation process. Finally, in Sec 3.2, we discuss how our second contribution, DualMDM, can increase the control and diversity of the intra-personal dynamics generated by pre-trained interaction models such as in2IN.

### 3.1. in2IN: Interaction diffusion model

The architecture of our interaction diffusion model (in2IN) is founded on the principle that interactions between two persons exhibit a commutative property [28], denoted as $\{x_a, x_b\}$, which is considered to be equivalent to $\{x_b, x_a\}$. Building on this concept, we introduce a Transformer-based diffusion model in a Siamese configuration [9]. Two copies of the diffusion model are made, sharing parameters. Each network is responsible for processing its respective noisy motion inputs, $\mathbf{x}_a^t$ and $\mathbf{x}_b^t$, and aims to produce the denoised versions, $\mathbf{x}_a^0$ and $\mathbf{x}_b^0$. We predict the $x_0$ directly [7, 45] as this allows us to use kinematic losses. Once the losses have been calculated, the motion is noised back to $x^{t-1}$ to become the input of the next denoising iteration.

Similarly to [28, 39], our diffusion model architecture (Fig. 2) has a multi-head self-attention module where it learns the intra-personal dynamics of the motion, and a multi-head cross-attention module that combines the self-attention output with the motion of the other individual in the interaction, thus modeling the inter-personal dynamics. We also condition the generation with adaptive normal-

Figure 2. **in2IN diffusion model.** Our proposed architecture consists of a Siamese Transformer that generates the denoised motion of each individual in the interaction ($x_a^0$ and $x_b^0$). In the first stage, a self-attention layer models the intra-personal dependencies using the encoded individual condition and noisy motion of each person ($x_a^t$ and $x_b^t$). In the second stage, a cross-attention module models the inter-personal dynamics using the encoded interaction description, the self-attention output, and the noisy motion from the other interacting person.

ization layers [30]. However, in contrast to previous approaches, we introduce different conditions for the different attention modules. For the self-attention module, where only the individual motion matters, we provide the specific textual description of the individual motion as conditioning. On the other hand, in the cross-attention module, where the whole interaction is important, we provide the interaction textual description as conditioning. This allows for a more precise control of the intra- and inter-personal dynamics.

**Multi-Weight Classifier-Free Guidance.** Our conditioning strategy for the diffusion model builds upon CFG, initially proposed by Ho *et al.* [22]. Generally, diffusion models have a significant dependency on CFG to generate high-quality samples. However, incorporating multiple conditions using CFG is not trivial. We address this by employing distinct weighting strategies for each condition. The equation representing our model's sampling function, denoted as $G_s(x^t, t, c)$, is as follows:

$$
\begin{aligned}
G_s\left(x^t, t, c\right) = {} & G\left(x^t, t, \emptyset\right) \\
& + w_c \cdot \left(G\left(x^t, t, c\right) - G\left(x^t, t, \emptyset\right)\right) \\
& + w_I \cdot \left(G\left(x^t, t, c_I\right) - G\left(x^t, t, \emptyset\right)\right) \\
& + w_i \cdot \left(G\left(x^t, t, c_i\right) - G\left(x^t, t, \emptyset\right)\right),
\end{aligned} \tag{1}
$$

where $G(x^t, t, \emptyset)$ is the unconditional output of the model, and $G(x^t, t, c)$, $G(x^t, t, c_I)$, and $G(x^t, t, c_i)$ denote the model outputs conditioned on the whole conditioning $c = \{c_I, c_i\}$, only the interaction, and only the individual, respectively. The weights $w_c$, $w_I$, and $w_i \in \mathbb{R}$ adjust the influence of each conditioned output relative to the unconditional baseline. A notable limitation of this approach is the necessity to perform quadruple sampling from the de-

noiser, as opposed to the dual sampling required in a conventional CFG methodology. In exchange, this method allows for more refined control over the generation process, ensuring that the model can effectively capture and express the nuances of both individual and interaction-specific conditions. If a weight is set to 0, then that particular conditioning is ignored during the generation process.

### 3.2. DualMDM: Model composition

In our second contribution, we propose a motion model composition technique that allows us to combine interactions generated by an interaction model with the motions generated by an individual motion prior trained with a single-person motion dataset. This method uses a single-person human motion prior to provide the generated human-human interactions with a higher diversity of intra-personal dynamics. This model composition technique is built on top of the method proposed in DiffusionBlending [36]:

$$
\begin{aligned}
G^{a,b}(x^t, t, c_a, c_b) = {} & G^a(x^t, t, c_a) \\
& + w \cdot (G^b(x^t, t, c_b) - G^a(x^t, t, c_a)),
\end{aligned} \tag{2}
$$

where $w \in \mathbb{R}$ is the blending weight, $G^a(x^t, t, c_a)$ and $G^b(x^t, t, c_b)$ are the outputs of the diffusion models $a$ and $b$, respectively. Since the original proposal was made to combine single-person diffusion models, we adapt the previous formula to our scenario:

$$
\begin{aligned}
G^{I,i}(x^t, t, c) = {} & G^I(x^t, t, c) \\
& + w \cdot (G^i(x^t, t, c_i) - G^I(x_t, t, c)),
\end{aligned} \tag{3}
$$

where $G^I(x^t, t, c)$ is the output of the interaction diffusion model and $G^i(x^t, t, c_i)$ is the output of the individual motion prior. By choosing $w$ to be constant, authors from

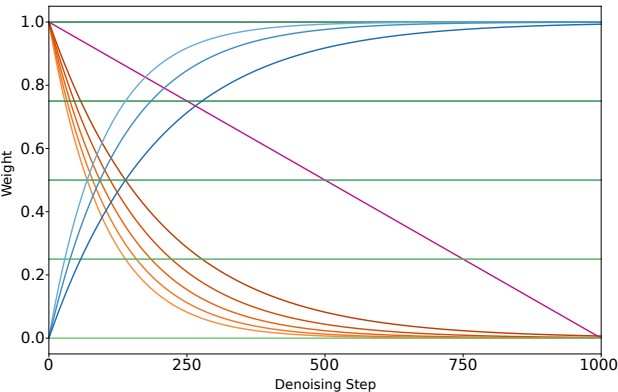

Figure 3. Different weights schedulers tested for DualMDM. **Oranges:** Exponential. **Blues:** Inverse Exponential. **Greens:** Constant. **Magenta:** Linear.

[36] assumed that the optimal blending weight is the same along the whole sampling process. However, in line with [24], we argue that the optimal blending weight might vary along the denoising chain, depending on the particularities of each scenario. To account for this, we propose to replace the constant $w$ with a weight scheduler $w(t)$ that parameterizes the blending weight used to combine the denoised motion from both models, making it variable on the sampling phase (Fig. 3). As a generalization of the DiffusionBlending technique, DualMDM is a more flexible and powerful strategy to combine two diffusion models.

## 4. Experimental Evaluation

### 4.1. Data

Our experiments are conducted with the InterHuman [28] and HumanML3D [17] datasets. InterHuman is the largest annotated interaction dataset in which each motion is represented as $x^i = \left[\mathbf{j}_g^p, \mathbf{j}_g^v, \mathbf{j}^r, \mathbf{c}^f\right]$, where $x^i$, the $i$-th motion timestep, encompasses joint positions $\mathbf{j}_g^p \in \mathbb{R}^{3N_j}$ and velocities $\mathbf{j}_g^v \in \mathbb{R}^{3N_j}$ in the world frame, $6D$ representation of local rotations $\mathbf{j}^r \in \mathbb{R}^{6N_j}$ in the root frame, and binary foot-ground contact features $\mathbf{c}^f \in \mathbb{R}^4$. $N$ is the number of joints. In our case $N = 22$. As InterHuman does not provide individual textual descriptions of the motions pertaining to the interaction, we have automatically generated them using LLMs.

InterHuman dataset is focused on providing a wide range of interactions rather than individual diversity in its motions. We have trained an individual motion prior with the HumanML3D dataset, which contains a much wider range of annotated individual motions. For compatibility purposes, we converted the HumanML3D motion representation to the one used in the InterHuman dataset. More details on the LLM-based generation of the individual descriptions and the implementation details of our individual

motion prior can be found in the Supplementary Material.

### 4.2. Evaluation Metrics

We utilize the evaluation metrics proposed in [17]. R-precision and Multimodal-Dist evaluate how semantically close the generated motions are to the input prompts. The FID score is used to measure the dissimilarity between the distributions of generated motions and the actual ground truth motions. Diversity is assessed to gauge the range of variation within the generated motion distribution, while MultiModality calculates the average variance for motions generated from a single text prompt. To compute these metrics, we need encoders that align the text and motion latent representation, which we borrow from [28].

None of the previous evaluation metrics assesses the alignment of the generated interactions with the individual descriptions. Due to the lack of ground-truth individual annotations, we cannot train single-person motion and text encoders for InterHuman. Therefore, we cannot reliably assess the individual alignment with the R-Prec, Multimodal-Dist, or FID metrics. We argue though that the interaction metrics are not only sensitive to the global quality of the interaction but also to the coherence of the intra-personal dynamics with the interaction context. If an interactant is *kicking a ball*, the *salute to each other* interaction is not coherent, and the generated motion will have low R-Prec. Thus, interaction metrics are indeed sensitive to wrong intra-personal dynamics in an interaction. What they do not capture are the intra-personal differences promoted by the usage of distinct individual descriptions. More specifically, the interaction generated with $\{c_{\mathrm{I}}=salute$ *to each other*, $c_{i_1}=c_{i_2}=wave$ *right hand*$\}$ will be different from the one generated with the same set with $c_{i_2}=bows$ *forward* instead. However, these differences might come 1) from the intrinsic diversity of the generative model, quantified by the MultiModality metric (i.e., different ways of waving right hand, and not bowing at all), or 2) from the extrinsic diversity caused by differences in the individual descriptions used, thus showing control capabilities over the generated intra-personal dynamics. With the motivation of quantifying the latter, we introduce a new evaluation metric called *Extrinsic Individual Diversity (EID)*.

**Extrinsic Individual Diversity (EID).** In order to assess the extrinsic diversity of the model, we need to disentangle it from the intrinsic one. To do so, we generate two empirical distributions that will serve as a proxy for quantifying the intrinsic diversity of 1) the ground-truth scenario, and 2) a synthetic scenario where the individual descriptions are randomly changed. In particular, for every set of interaction and individual descriptions $\{c_{\mathrm{I}}, c_{i_1}, c_{i_2}\}$ in the dataset, we proceed as follows: 1) we build $D_{\mathrm{GT}}$ as the set of $N$ motions generated with $\{c_{\mathrm{I}}, c_{i_1}, c_{i_2}\}$, and 2) we build $D_{\mathrm{rand}}$ as the set of $N$ motions generated randomly replacing $c_{i_1}$

and $c_{i_2}$ with other individual descriptions from the dataset. Then, we define the *EID* as the Wasserstein distance between $D_{GT}$ and $D_{rand}$. A higher distance means more influence of the individual descriptions on the diversity of the generated motions, arguably leading to higher control on the intra-personal dynamics of the interaction. This metric can be combined with others such as the R-Precision and FID to analyze the trade-off between individual diversity and interaction quality and fidelity.

In our experiments, we set $N=32$. To quantify the additional extrinsic diversity provided by the DualMDM technique, we build $D_{GT}$ with in2IN and $D_{rand}$ with in2IN combined with the DualMDM.

## 4.3. Implementation Details

Our in2IN models consist of 8 consecutive multi-head attention layers with a latent dimension of 1024 and 8 heads. We utilize a frozen CLIP-ViT$L/14$ model [35] as our text encoder. We set the number of diffusion timesteps to 1,000 and employ a cosine noise schedule [31]. During inference, we use DDIM sampling [38] with $\eta = 0$ and 50 timesteps, and our proposed multi-weight CFG variation. To enable the latter, 10% of the CLIP embeddings are randomly set to zero during training.

All models have been trained using the AdamW optimizer [29] with betas of $(0.9, 0.999)$, weight decay of $2 \times 10^{-5}$, maximum learning rate of $10^{-4}$, and a cosine learning rate schedule with an initial 10-epoch linear warmup period. They have been trained using the L2 loss and, thanks to the use of the $x_0$ parameterization, kinematic losses have also been used. These include the foot contact and the velocity losses from the MDM framework [41], and the bone length, the masked joint distance map, and the relative orientation losses suggested in InterGen [28]. Additionally, we have used the kinematic loss scheduler from InterGen. All models have been trained for 2,000 epochs with a batch size of 64 with 16-bit mixed precision. Two Nvidia 3090 GPUs have been required for the span of 5 days.

**DualMDM schedulers.** We test these functions:

$$
\begin{aligned}
&\text{constant, or} && w(t) = \lambda \\
&\text{linear, or} && w(t) = t/T \\
&\text{exponential, or} && w(t) = e^{-\lambda \cdot (T-t)}, \\
&\text{inverse exponential, or} && w(t) = 1 - e^{-\lambda \cdot (T-t)},
\end{aligned}
\tag{4}
$$

where $t$ is the actual denoising step, $T$ is the total number of denoising steps, and $\lambda$ is the parameter that determines the slope of our scheduler function. We visualize them in Fig. 3.

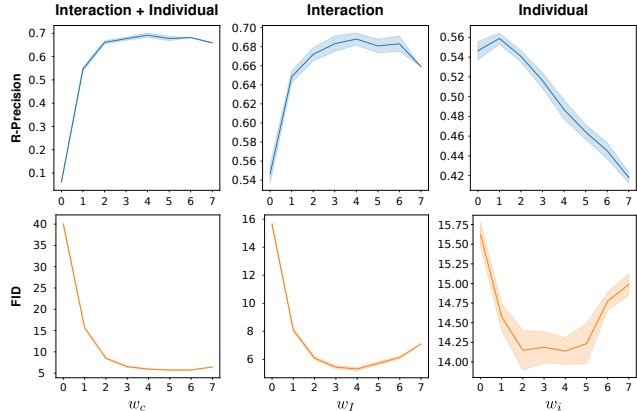

Figure 4. Comparison of **R-Precision** and **FID** for the different weights on the Multi-Weight CFG tested in isolation. Each column represents a different weight $(w_c, w_I, w_i)$. $w_c$ has been tested with $w_I=w_i=0$. $w_I$ and $w_i$ have been tested with $w_c=1$, and the other weight set to 0.

## 4.4. Quantitative Analysis

### 4.4.1 in2IN: Interaction Generation

Tab. 1 shows the quantitative evaluation of our in2IN architecture with respect to the previously existing methods evaluated on the InterHuman dataset. It can be observed that by using individual information we have been able to obtain better results than all previous methods. As might reasonably be anticipated, the additional information used only by in2IN in form of LLM-generated individual descriptions reduces the spectrum of valid motions fulfilling the interaction description, which reflects as a lower MultiModality.

With respect to the Multi-Weight CFG, we evaluate the isolated effect of each weight on the evaluation metrics in Fig. 4. As can be observed, for weights $w_c$ and $w_I$, 4 is the best weight individually. On the other hand, for weight $w_i$, 2 is the best weight. More than that turns into a decrement in performance. We find the best combination with a grid search in a validation subset: $w_c=3$, $w_I=3$, and $w_i=1$.

### 4.4.2 DualMDM: Individual Diversity

In Tab. 2, the EID metric is compared with the R-Precision and FID using different schedulers in our DualMDM method. In general, we can observe that in all the schedulers, the ones that assign more weight to the individual model obtain higher individual diversity, in exchange for a lower interaction quality. While a constant scheduler with $\lambda=0.25$ seems to achieve good quantitative values, we can observe that the exponential weight scheduler with $\lambda=0.00875$ provides a better trade-off between individual diversity and interaction quality. This is fundamental, as we want to have high intra-personal diversity while keeping the inter-personal coherence. We hypothesize that the good

| Methods | R Precision ↑ | | | FID ↓ | MM Dist ↓ | Diversity → | MModality ↑ |
| --- | --- | --- | --- | --- | --- | --- | --- |
| | Top 1 | Top 2 | Top 3 | | | | |
| Ground Truth | $0.452^{\pm.008}$ | $0.610^{\pm.009}$ | $0.701^{\pm.008}$ | $0.273^{\pm.007}$ | $3.755^{\pm.008}$ | $7.948^{\pm.064}$ | - |
| TEMOS [33] | $0.224^{\pm.010}$ | $0.316^{\pm.013}$ | $0.450^{\pm.018}$ | $17.375^{\pm.043}$ | $6.342^{\pm.015}$ | $6.939^{\pm.071}$ | $0.535^{\pm.014}$ |
| T2M[17] | $0.238^{\pm.012}$ | $0.325^{\pm.010}$ | $0.464^{\pm.014}$ | $13.769^{\pm.072}$ | $5.731^{\pm.013}$ | $7.046^{\pm.022}$ | $1.387^{\pm.076}$ |
| MDM [41] | $0.153^{\pm.012}$ | $0.260^{\pm.009}$ | $0.339^{\pm.012}$ | $9.167^{\pm.056}$ | $7.125^{\pm.018}$ | $7.602^{\pm.045}$ | $\mathbf{2.35}^{\pm.080}$ |
| ComMDM [36] | $0.223^{\pm.009}$ | $0.334^{\pm.008}$ | $0.466^{\pm.010}$ | $7.069^{\pm.054}$ | $6.212^{\pm.021}$ | $7.244^{\pm.038}$ | $1.822^{\pm.052}$ |
| InterGen [28] | $0.371^{\pm.010}$ | $0.515^{\pm.012}$ | $0.624^{\pm.010}$ | $5.918^{\pm.079}$ | $5.108^{\pm.014}$ | $7.387^{\pm.029}$ | $\underline{2.141}^{\pm.063}$ |
| MoMat-MoGen [11] | $\underline{0.449}^{\pm.004}$ | $\underline{0.591}^{\pm.003}$ | $\underline{0.666}^{\pm.004}$ | $5.674^{\pm.085}$ | $\mathbf{3.790}^{\pm.001}$ | $8.021^{\pm.035}$ | $1.295^{\pm.023}$ |
| in2IN* | $0.425^{\pm0.008}$ | $0.576^{\pm0.008}$ | $0.662^{\pm0.009}$ | $\underline{5.535}^{\pm0.120}$ | $3.803^{\pm0.002}$ | $\mathbf{7.953}^{\pm0.047}$ | $1.215^{\pm0.023}$ |
| in2IN | $\mathbf{0.455}^{\pm0.004}$ | $\mathbf{0.611}^{\pm0.005}$ | $\mathbf{0.687}^{\pm0.005}$ | $\mathbf{5.177}^{\pm0.103}$ | $\mathbf{3.790}^{\pm0.002}$ | $\underline{7.940}^{\pm0.030}$ | $1.061^{\pm0.038}$ |

Table 1. Comparison of our model (in2IN) to the state of the art in human-human interaction motion generation on the InterHuman dataset. *in2IN model only using $w_I$ (conditioning only on the interaction during sampling). All evaluations have been executed 10 times to elude the randomness of the generation $\pm$ indicates the 95% confidence interval. We highlight the **best** and the second best results.

| Scheduler $\lambda$ | R Precision ↑ | FID ↓ | EID ↑ |
| --- | --- | --- | --- |
| 0.00 | $\mathbf{0.687}^{\pm.005}$ | $\mathbf{5.177}^{\pm.103}$ | $1.238^{\pm.011}$ |
| 0.25 | $0.577^{\pm.004}$ | $33.75^{\pm.293}$ | $1.516^{\pm.005}$ |
| 0.50 | $0.383^{\pm.006}$ | $91.99^{\pm.000}$ | $1.972^{\pm.018}$ |
| 0.75 | $0.218^{\pm.016}$ | $127.8^{\pm.691}$ | $\mathbf{2.188}^{\pm.010}$ |
| 1.00 | $0.094^{\pm.004}$ | $130.4^{\pm.226}$ | $2.118^{\pm.010}$ |
| 0.0100 | $\mathbf{0.589}^{\pm.006}$ | $\mathbf{19.76}^{\pm.232}$ | $1.461^{\pm.007}$ |
| 0.00875 | $0.574^{\pm.003}$ | $22.86^{\pm.190}$ | $1.492^{\pm.006}$ |
| 0.0075 | $0.565^{\pm.007}$ | $26.20^{\pm.129}$ | $1.534^{\pm.013}$ |
| 0.00625 | $0.530^{\pm.013}$ | $31.23^{\pm.211}$ | $1.596^{\pm.009}$ |
| 0.0050 | $0.500^{\pm.007}$ | $39.36^{\pm.301}$ | $\mathbf{1.680}^{\pm.004}$ |
| 0.0100 | $0.232^{\pm.006}$ | $114.3^{\pm.433}$ | $\mathbf{2.140}^{\pm.013}$ |
| 0.0075 | $0.251^{\pm.004}$ | $111.1^{\pm.316}$ | $2.115^{\pm.008}$ |
| 0.0050 | $\mathbf{0.282}^{\pm.006}$ | $\mathbf{106.8}^{\pm.386}$ | $2.088^{\pm.009}$ |
| - | $\mathbf{0.235}^{\pm.005}$ | $\mathbf{98.27}^{\pm.528}$ | $2.118^{\pm.010}$ |

Table 2. Table comparing the Extrinsic Individual Diversity (EID) and interaction metrics of different weight schedulers. **Oranges:** Exponential. **Blues:** Inverse Exponential. **Greens:** Constant. **Magenta:** Linear. **Bold** represents the best value for each scheduler.

trade-off acquired by the exponential schedule is due to the fact that the intra-relationships of the motion (provided by the individual motion prior) are much more important during the early stages of denoising. However, as the sampling advances, the inter-relationships of the motions interaction become more relevant. Also, when the individual model is used during the later stages of denoising, it deteriorates the denoised inter-personal dynamics. On the contrary, if the weight on this individual prior is gradually reduced, the interaction model is able to recover these dynamics in the later stages of the denoising. In Sec. 4.5, we validate some of these hypothesis by means of a qualitative analysis.

### 4.5. Qualitative Analysis

As depicted in Fig. 5 and Fig. 6, our in2IN model can generate more realistic interactions aligned with the textual description. Upon qualitative evaluation, our model consistently outperforms InterGen across various scenarios. Fig. 7 illustrates the effect of the different weighting strategies for our DualMDM motion composition method. It can be observed how the exponential scheduler provides more coherent results, preserving the interaction semantics while generating individual motions that match the individual descriptions, yielding a superior fine-grained control. While a constant scheduler might quantitatively provide decent results, the qualitative evaluation demonstrates the superiority of the exponential scheduler. For the constant schedulers, we notice that increasing the weight assigned to the individual prior leads to a degradation of the inter-personal dynamics, particularly concerning trajectories and orientations. As a limitation of the exponential scheduler, we can observe that the $\lambda$ value selected for each case is critical and might not be the same for all compositions. The selection of this value will depend on the specific characteristics of the interaction and individual motions that we want to combine. More visualizations supporting these observations can be found in the Supplementary Material.

## 5. Conclusion

We presented in2IN, an interaction diffusion model that leverages both interaction and individual textual descriptions to generate better inter- and intra-personal dynamics in the human-human motion interaction generation. With a more precise conditioning, in2IN has become the new state of the art in the InterHuman dataset. We also introduced DualMDM, a motion model composition technique that injects the single-person dynamics learned by a pre-trained individual motion prior into the generated interactions. As a result, combining in2IN with DualMDM provides better control over the intra-personal dynamics of the interaction.

**Limitations and Future work.** One of our main reasons to propose DualMDM is that the optimal strategy for combining the outputs of the individual and the interaction models change along the sampling process. However, we observed in Sec. 4.5 that these dynamics vary as well depending on the descriptions, or even on the stochasticity of the generation itself. Future work includes exploring better blending strategies for which the user does not need to define any scheduler parameter.

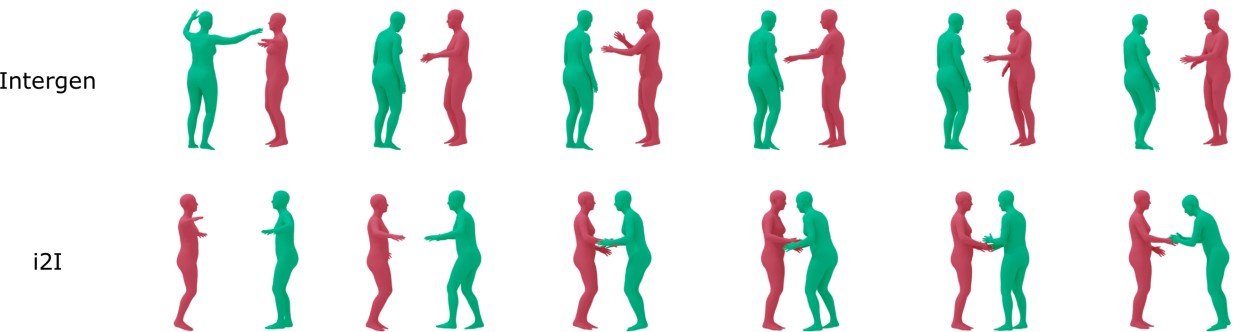

Figure 5. **Interaction Description:** The two guys meet, grip each other's hand, and nod in agreement. The X-axis represents time.

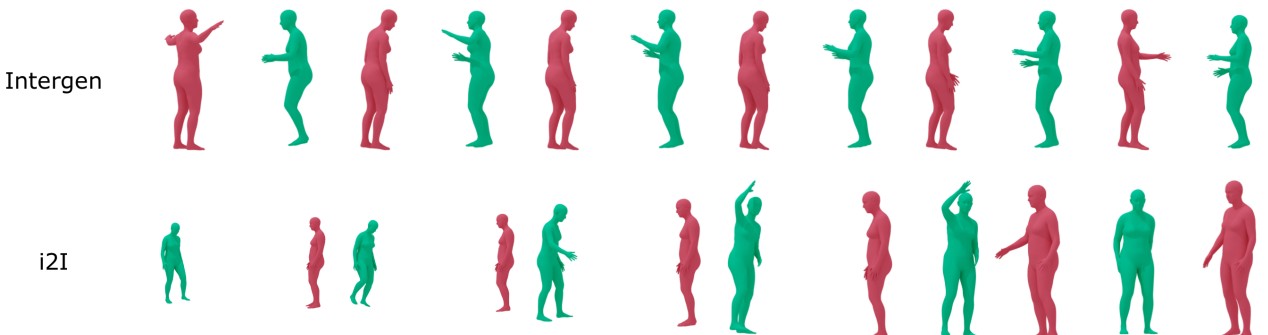

Figure 6. **Interaction Description:** One person spots the other person on the street, lifts the right hand to greet, and the other person glances towards one person. The X-axis represents time.

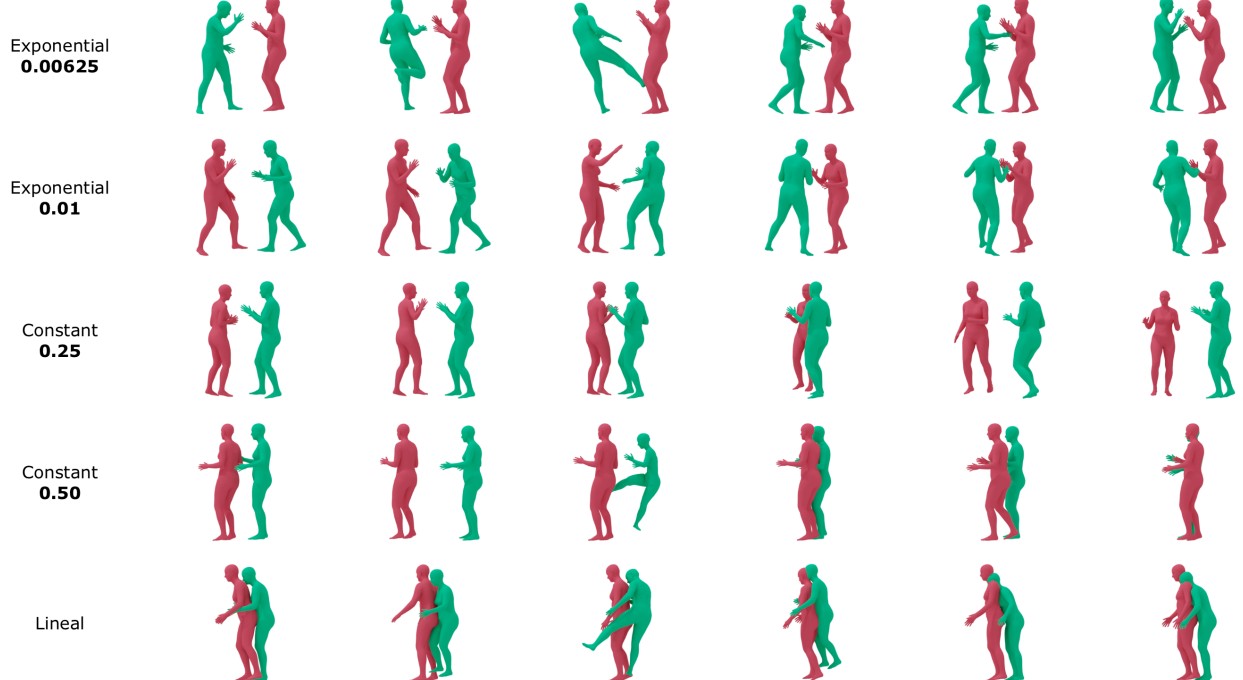

Figure 7. **Interaction Description:** Two persons are in an intense boxing match. **Individual Description #1:** An individual throws a kick with his right leg. **Individual Description #2:** An individual is boxing. The X-axis represents time.

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
