# OpenReview forum: "in2IN: Leveraging individual Information to Generate Human INteractions"
_thecvf.com/CVPR/2024/Workshop/HuMoGen — CVPR 2024 Workshop HuMoGen Submission_

### Official Review · Reviewer_qbkr · 2024-03-29
**Enhancing Motion Generation through Data Integration: A Review with Recommendations for Clarity and Methodological Coherence**

**Rating:** 3
**Confidence:** 4

**Review:**

The authors propose an intriguing approach by integrating individual and interaction datasets to enhance motion generation control for each individual and their interactions. This concept is solid and great and represents a significant advancement in the field. However, the articulation of the paper fluctuates between clarity and ambiguity, impacting the delivery of its contributions.

The structure of the paper is generally satisfactory. Nonetheless, certain aspects require elucidation:

1. Methodological Clarity: The introduction of CFG into both interaction and individual diffusion models is innovative. However, the relationship between equation (1) and equation (3) remains ambiguous. It appears as though these equations originate from disparate methodologies, lacking a coherent integration. This aspect suggests a potential oversight in the methodological framework, which warrants clarification for a unified understanding.

2. Experimental Results and Analysis: The experimental section outlines the influence of the scheduler on the outcomes effectively. Yet, the identical figures in the concluding row of Table 1 and the opening row of Table 2, coupled with the \lambda value set to 0.0, introduce confusion. It implies that when the individual has no effect, the model performs better in terms of r-precision and FID. I believe it is important to conduct further experiments to investigate this phenomenon.

3. Data Utilization and Result Interpretation: The presentation of the results does not fully address the anticipated improvements with the incorporation of more comprehensive data sources. The transparency in reporting Table 2 is commendable; however, the expected enhancement in results with additional data  (both two-person and single-person datasets)  was not observed, lacking a satisfactory explanation for this discrepancy.

4. Writing and Logical Flow: The manuscript's logical flow, particularly in the section spanning lines 342 to 369, could benefit from refinement. The primary concept, though discernible, is presented in a manner that obfuscates its significance. Additionally, the use of certain expressions, such as "though" in line 348, introduces unnecessary confusion. Refining this segment could substantially enhance the comprehension of the EID concept, which is pivotal to the paper.

The paper introduces a compelling concept, yet struggles with clarity, experimental validation, and exposition, leaving me feeling complicated about its current form. Addressing these issues could significantly elevate the manuscript, highlighting its innovative contributions more effectively. This work has undeniable potential; with focused improvements, it could truly impact the field.

---

### Official Review · Reviewer_4Fyf · 2024-04-01
**The paper presents very interesting concepts but lack explanations.**

**Rating:** 4
**Confidence:** 5

**Review:**

The authors present intriguing concepts and innovative contributions, including:

* The novel diffusion model architecture, which generates two motions symmetrically conditioned on both individual prompts and general interaction prompts, allowing for greater diversity in each character's behavior, following its specific prompt rather than a general one.
* A new technique is demonstrated to guide generation independently towards each condition via multiple CFG.
* The novel motion composition technique - DualMDM, enables the use of a single motion diffusion model's prior.
* The impressive improvment quantitativly on almost all metrices using the new methods mentioned above.
* The qualitative result outperform current state of the art.

yet lacks comprehensive explanations regarding their workings and the impact of their novelty on the achieved results.
* Regarding the architecture of the model, how does information flows between the two models during generation? how is one inference aware of the generated motion from the second inference?
* during inference, what is the latency of generating one motion? Since the duplication of the model times the 4 CFG and the single motion prior model may result in a lot of resources required for a single generation.
* In Table 1, the authors did not quantify the contribution of DualMDM to the obtained results.
* Figure 2 depicts both individual embedding and interaction embedding entering the CLIP and linear layer as two inputs, resulting in an output dependent on both. However, the textual explanation suggests that they enter separately into the CLIP model and linear layer.

If the papers will be accepted, I ask the Authors to explain the list of points lacking of information in the camera ready copy.

After considering these factors, I incline for a weak accpet.

---

### Meta-Review · Area_Chair_UZJL · 2024-04-05

**Recommendation:** Accept

**Metareview:**

The paper addresses the task of synthesizing human interaction motion from text.

Pros:
* Multiple novel components
* Quantitative and Qualitative results outperform current work

Cons:
* Some details are missing in the paper
* During training, the decomposition of the textural description into two individual sentences does not necessarily match the ground truth motions. This issue should be specified as a limitation

**Guidance to authors:**
* Please resolve concerns raised by the authors, particularly ones related to missing details
* Please ensure that the new dataset (i.e., individual text descriptions) is made publicly available

---

### Decision · Program_Chairs · 2024-04-06

**Decision:**

Accept

**Comment:**

The paper will be published as part of the official CVPR workshop proceedings upon submission of the camera-ready version.